

# Ecology and life history of *Meta bourneti* (Araneae: Tetragnathidae) from Monte Albo (Sardinia, Italy)

Enrico Lunghi

Department of Biogeography, Trier University, Trier, Germany
Sezione di Zoologia ''La Specola'', Museo di Storia Naturale dell'Università di Firenze, Firenze, Italy
Natural Oasis, Prato, Prato, Italy

## ABSTRACT

The orb-weaver spider *Meta bourneti* Simon 1922 (Araneae: Tetragnathidae) is one of the most common cave predators occurring in the Mediterranean basin. Although the congeneric *M. menardi* represented the model species in several studies, our knowledge of *M. bourneti* is only founded on observations performed on a handful of populations. In this study *M. bourneti* spiders were studied in caves of Monte Albo (Sardinia, Italy) over a year. Generalized Linear Mixed Models were used to analyze spider occupancy inside cave environments, as well as spider abundance. Analyses on *M. bourneti* occupancy and abundance were also repeated for adults and juveniles separately. Generalized Linear Models, were used to weight species absence based on its detection probability. Linear Mixed Models were used to detect possible divergences in subterranean spatial use between adult and juvenile spiders. Although widespread on the mountain, *M. bourneti* generally showed low density and low detection probability. Most of the individuals observed were juveniles. The spiders generally occupied cave sectors with high ceilings that were deep enough to show particular microclimatic features. Adults tended to occupy less illuminated areas than juveniles, while the latter were more frequently found in sectors showing high humidity. The abundance of *M. bourneti* was strongly related to high humidity and the presence of two troglophile species, *Hydromantes flavus* Wake, Salvador & Alonso-Zarazaga, 2005 (Amphibia: Caudata) and *Oxychilus oppressus* (Shuttleworth, 1877) (Gastropoda: Panpulmonata). The abundance of juveniles was related to sector temperature and humidity, the presence of *H. flavus* and *O. oppressus* and to morphological sector features. However, when only adults were considered, no significant relationships were found. Adult and juvenile spiders did not differ in their spatial distribution inside the caves studied, but a seasonal distribution of the species along cave walls was observed. Microclimate was one of the most important features affecting both the presence and abundance of *M. bourneti* in subterranean environments. Individuals tended to occupy lower heights during hot seasons.

Corresponding author
Enrico Lunghi, enrico.arti@gmail.com

## INTRODUCTION

Subterranean environments, from shallow cracks and burrows to the deepest karst systems, are peculiar habitats showing a characteristic combination of environmental features. They generally show little or no light, high air humidity and a relatively stable temperature resembling the mean annual temperature occurring in outdoor surrounding areas (*Culver & Pipan, 2009*; *Smithson, 1991*). Subterranean microclimate is generally shaped by the influence of external climate which, through openings connecting subterranean environments with outer ones, spread in and contribute to creating different microhabitats (*Badino, 2004*; *Badino, 2010*; *Campbell Grant, Lowe & Fagan, 2007*; *Lunghi, Manenti & Ficetola, 2015*). The most evident result is the formation of three different macro-ecological zones (*Culver & White, 2005*). The zone adjacent to the connection with the outdoor is the most affected by external influences. Indeed, the microclimate of this area generally resembles environmental conditions occurring in surrounding outdoor areas. In the "twilight zone" external influences are weaker and incoming light is generally low. Finally, there is the deep zone, where incoming light is absent and microclimatic features are the most stable.

Subterranean environments house a rich biodiversity of species that display unique and peculiar adaptations to the different ecological zones (*Romero, 2011*). A species' degree of association to subterranean conditions is the basis for the general ecological classification used to distinguish between different groups of cave-dwelling organisms (*Christiansen, 1962*; *Novak et al., 2012*; *Pavan, 1944*; *Sket, 2008*). Several additional descriptors are used to classify cave animals (*Trajano & De Carvalho, 2017*). The most specialized are called troglobites, species closely connected to the deep areas of subterranean environments. Troglobites often show specific adaptations, such as depigmentation, anophthalmia, elongation of appendages, and a reduction in metabolic rates (*Aspiras et al., 2012*; *Bilandžija et al., 2013*; *Biswas, 2009*; *Hervant, Mathieu & Durand, 2000*). In contrast, troglophiles can exploit both epigan and hypogean environments and their adaptations to cave life are reduced or even absent (*Di Russo et al., 1999*; *Fenolio et al., 2006*; *Lunghi, Manenti & Ficetola, 2017*). Trogloxenes are epigean species accidentally found in the shallowest part of subterranean environments. This classification, however, is viewed too strict (*Lunghi, Manenti & Ficetola, 2014*; *Romero, 2009*), as species usually thought to be accidental are indeed potential residents playing an important role throughout the entire ecosystem (*Lunghi et al., 2018a*; *Manenti, Lunghi & Ficetola, 2017*; *Manenti, Siesa & Ficetola, 2013*).

Despite an increasing interest in subterranean ecological spaces and their related biodiversity that has occurred in the last decades (*Culver & Pipan, 2009*; *Culver & Pipan, 2014*; *Juan et al., 2010*; *Romero, 2009*), our current knowledge of cave-dwelling species is incomplete. For example, there is the case of the troglophile orb-weaving spider *Meta bourneti* Simon 1922 (Araneae, Tetragnathidae). *Meta* spiders are among the most common predators in cave environments (*Mammola & Isaia, 2017b*; *Mammola, Piano & Isaia, 2016*; *Manenti, Lunghi & Ficetola, 2015*; *Pastorelli & Laghi, 2006*). These spiders show an interesting complex life history. During their early life stages they are phototaxic and disperse in outdoor environments, while during the adult phase they become photophobic

and inhabit subterranean environments, where they reproduce (*Chiavazzo et al., 2015*; *Smithers, 2005b*; *Smithers & Smith, 1998*; but see also Fig. 6 in *Mammola & Isaia, 2014*). *Meta* spiders are at the apex of the subterranean food-chain, preying on several species using both web and active hunting (*Lunghi, Manenti & Ficetola, 2017*; *Mammola & Isaia, 2014*; *Novak et al., 2010*; *Pastorelli & Laghi, 2006*; *Smithers, 2005a*; *Tercafs, 1972*). However, young spiders are potential prey of other cave predators (*Lunghi et al., 2018b*).

In Europe and the Mediterranean basin area, two species of *Meta* spiders are commonly observed, *M. menardi* and *M. bourneti* (*Fernández-Pérez, Castro & Prieto, 2014*; *Fritzén & Koponen, 2011*; *Mammola & Isaia, 2014*; *Nentwig et al., 2018*). Although the former is the subject of several studies (*Ecker & Moritz, 1992*; *Hörweg, Blick & Zaenker, 2012*; *Lunghi, Manenti & Ficetola, 2017*; *Mammola, Piano & Isaia, 2016*; *Manenti, Lunghi & Ficetola, 2015*), research on *M. bourneti* is very limited (*Boissin, 1973*; *Mammola, 2017*; *Mammola & Isaia, 2017a*). In a recent study, *Mammola & Isaia (2014)* studied the distribution and abundance of *M. menardi* and *M. bourneti* in six caves located in the north-west of Italy. Although they confirmed the previously hypothesized similarities in habitat selection between the two cave-dwelling *Meta* spiders (*Gasparo & Thaler, 1999*), *M. bourneti* was present at warmer temperatures. In addition, it displayed a shift in its life cycle compared to the congeneric *M. menardi*, which likely resulted from competition between the two species (*Mammola & Isaia, 2014*).

The present study provides the first report of the ecology and life history of *M. bourneti* populations from Sardinia (Italy). In this area the congeneric *M. menardi* is not present and thus, no potential interspecific interactions limit habitat selection of *M. bourneti* (*Mammola & Isaia, 2014*; *Nentwig et al., 2018*). This study aimed to produce information related to: (i) Improve our understanding of the effect of abiotic and biotic factors on both the occupancy and abundance of *M. bourneti* in subterranean environments, (ii) document the spatial distribution of these spiders within caves, (iii) identify differences between life stages (juveniles vs adults), and (iv) gather and summarize information on the life history of the species.

## MATERIALS & METHODS

### Dataset

The analyzed dataset focuses on *M. bourneti* observed in caves from the Monte Albo (north-east Sardinia, Italy) (Fig. 1; Table S1). Data were collected from seven different caves. However, *M. bourneti* was not observed in one of the caves and the cave was not included in the analysis. Surveys were performed seasonally, from autumn 2015 to summer 2016, thus covering a full year. Two samplings, 1–7 days apart, were conducted each season. Using a meter tape, inner cave environments were divided horizontally into 3 m sections (hereafter, sector), to collect fine-scale data on both cave morphology and microclimate, as well as on the occurrence of other cave-dwelling species (*Ficetola, Pennati & Manenti, 2012*; *Lunghi, Manenti & Ficetola, 2017*). Caves were explored entirely or up to the point reachable without speleological equipment. Within each cave sector the following abiotic data were recorded: maximum height and width, wall heterogeneity, mean temperature

(°C), humidity (%) and illuminance (lux). Maximum height and width were recorded at the end of each sector using a laser meter (Anself RZE-70, accuracy 2 mm). At each of these sampling points wall heterogeneity (i.e., presence of wall protuberance) was measured by placing a one-meter length of string along the cave wall at each of the sampling points between 0.5–2 m of height. The string was unrolled vertically following the shape of the cave wall, and the linear distance between the two string extremities (measured with a meter tape) quantified the smoothness of the wall (*Ficetola, Pennati & Manenti, 2012*; *Lunghi, Manenti & Ficetola, 2014*). During each survey, inner microclimatic data were recorded using a Lafayette TDP92 thermo-hygrometer (accuracy: 0.1 °C and 0.1%). At the end of each cave sector, the mean air temperature and humidity were estimated by merging data recorded in two different points: at ground level and at 2.5 m of height (or at the ceiling if sector height was lower). Microclimatic data were recorded paying attention to limit operator influence (*Lopes Ferreira et al., 2015*). At the same point, the maximum and minimum incident light using a Velleman DVM1300 light meter (minimum recordable light: 0.1 lux) was also measured. A standardized survey method (7.5 min/sector) was used to collect data on the presence of six cave-dwelling species: *M. bourneti*, *Hydromantes flavus* (Wake et al., 2005) (Amphibia: Caudata), *Metellina merianae* Scopoli, 1763 (Arachnida: Araneae), *Tegenaria* sp. Latreille, 1804 (Arachnida: Araneae), *Oxychilus oppressus* (Shuttleworth, 1877) (Gastropoda: Panpulmonata) and *Limonia nubeculosa* Meigen, 1804 (Insecta: Diptera). These species likely interact with *Meta* spiders, as they represent both potential prey and predators (*Lunghi et al., 2018b*; *Manenti, Lunghi & Ficetola, 2015*; *Novak et al., 2010*). *Meta* spiders were also counted and ascribed to two different categories on the basis of body size (prosoma + opisthosoma): adults with fully developed pedipalps (body size ≥ 10 mm) and juveniles (body size < 10 mm) (*Bellmann, 2011*; *Mammola & Isaia, 2014*; *Nentwig et al., 2018*). The number of observed cocoons was also recorded.

## Data analyses

The following analyses were performed in the open source statistical computing program R (*R Core Team, 2016*). Analyses on detection probability, species-habitat association and abundance were performed three times, one for each group studied (all individuals, adults only and juveniles only). Data for modeling species occurrence and abundance, was only related to surveys in which microclimatic features were recorded (cave surveys = 31, $N$ of spiders = 110).

### Detection probability

Cave spiders are among the species showing imperfect detection: a species is present when it is observed, but a lack of observation does not mean its true absence (*MacKenzie et al., 2006*). The detection probability of *M. bourneti* was estimated on the basis of twenty-seven pairs of cave surveys (i.e., 624 pairs of cave sectors) performed during each season with a gap ≤ 7 days (R package unmarked; *Fiske & Chandler, 2011*), a prerequisite for population closure (i.e., no immigration or emigration occurs; *MacKenzie et al., 2006*). Three possible covariates influencing spider detection were considered: the depth of the cave sector (hereafter, depth), the season and the wall heterogeneity. Four models were built, one

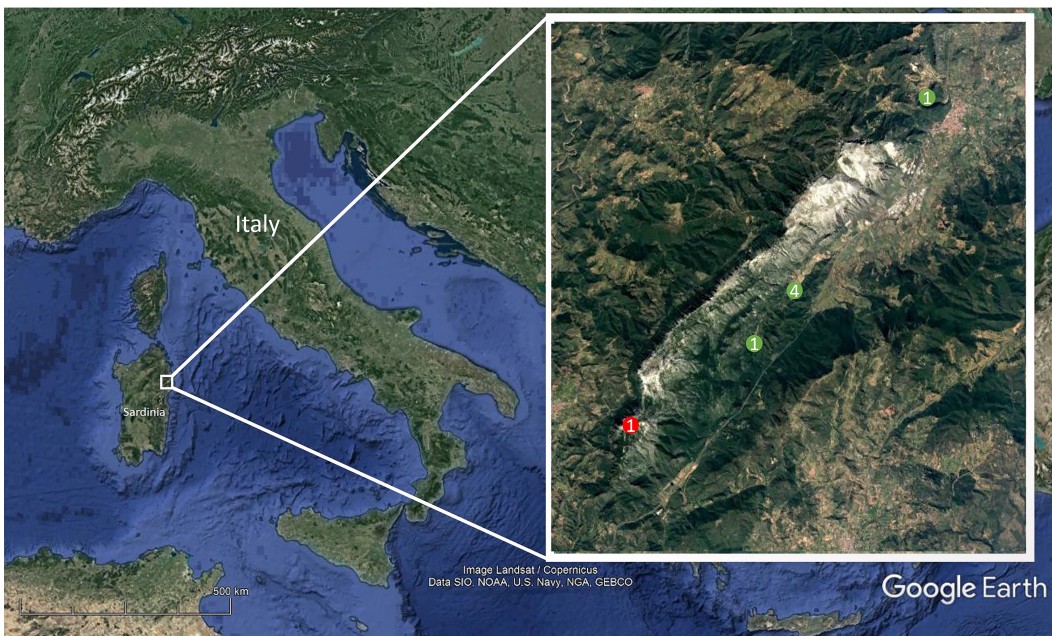

**Figure 1** **Map of the surveyed area.** The map shows the surveyed caves in Monte Albo (Sardinia, Italy). Green circles (and the respective number) indicate the caves in which was observed *Meta bourneti* spiders; the red circle indicates the cave in which the species was not observed. Map credit: Google Earth (Image Landsat/Copernicus; Data SIO, NOAA, US Navy, NGA, GEBCO).

for each covariate and one with none (i.e., the null model), and then ranked following the Akaike's Information Criterion (AIC); the one with the lowest AIC value was used to estimate detection probability (*Burnham & Anderson, 2002*; *Burnham, Anderson & Huyvaert, 2011*).

### Analyses on species occurrence

Binomial Generalized Linear Mixed Models (GLMM) (R packages lme4, lmerTest, MASS, MuMIn; *Bartoń, 2016*; *Douglas et al., 2015*; *Kuznetsova, Brockhoff & Christensen, 2016*; *Venables & Ripley, 2002*) were used to assess the relationship between *Meta* spiders and the abiotic features characterizing the cave environments. The presence/absence of the spiders was used as a dependent variable, while sector morphology (height, width and wall heterogeneity) and microclimatic features (temperature, humidity and illuminance) were used as independent variables. To evaluate whether spiders' preferences change through the year, the interaction between season and each of the considered microclimatic features was also included as an independent variable. Sector and cave identity were used as random factors. For each studied group, GLMMs were built using all possible combinations of independent variables; such models were then ranked following the Akaike's Information Criterion corrected for small sample size (AICc) (*Fang, 2011*). The model showing the lowest AICc value was considered the best model. Following the recommendations of *Richards, Whittingham & Stephens (2011)*, models representing more complicated versions of those with a lower AIC value and nested models were not considered as candidate
models. The likelihood ratio test was used to assess the significance of variables included in the best AICc models. Before analyses, humidity was angular-transformed and illuminance log-transformed, to improve linearity.

Considering a potential variation in species-habitat association over time (*Lunghi, Manenti & Ficetola, 2015*; *Lunghi, Manenti & Ficetola, 2017*) and an overall low detection probability estimated for these spiders, the robustness of the previous analyses was tested using a method that allows weighting the species absence on the basis of its detection probability: the General Linear Models (GLM) (*Gómez-Rodríguez et al., 2012*). Adding random factors to this analysis is not possible, hence the cave identity was included as a fixed factor. Following the same procedure described above, for each species all possible GLMs were built and ranked following AICc. The significance of variables included in the best AICc model was tested using the likelihood ratio test (*Bolker et al., 2008*).

GLM analysis was repeated for each group including depth as a further independent variable; as for some groups the best AICc model estimating detection probability included sector depth.

### Analyses of species abundance

The relationship between abundance of *M. bourneti* and both microclimatic and biotic recorded parameters was examined using GLMMs. The observed abundance of spiders was used as a dependent variable, as it represents an index of true abundance (*Barke et al., 2017*). Season, along with both microclimatic (temperature, humidity and illuminance) and biotic (presence/absence of the five considered species) features, were included as independent variables, while sector and cave identity were included as random factors. The significance of variables was tested with a Likelihood ratio test.

### Analyses on spatial distribution

Two Linear Mixed Models (LMM) (R package nlme; *Pinheiro et al., 2016*) were used to test whether adult and juvenile *M. bourneti* show divergences in the spatial use of subterranean environments; spiders' age class (adult/juveniles) and season were used as independent factors, and both sector and cave identity as random factors. The two dependent variables were the distance from the cave entrance and the height above cave floor, respectively. The dataset used in this analysis is shown in Table S2.

## RESULTS

Overall, a total of 182 observations of *M. bourneti* (64 adults and 118 juveniles) were performed within the caves studied (mean $\pm$ SE = 30.33 $\pm$ 16.49 per cave). Observations of spiders were highest in spring (3.17 spiders/visit), followed by winter (2.92 spiders/visit), summer (2.67 spiders/visit) and autumn (1.92 spiders/visit) (Fig. 2). Of 1,958 cave sector surveys, spiders were observed on 155 occasions, with generally one spider occupying the cave sector (132) (Table S2). Occupied cave sectors showed the following microclimatic conditions: mean temperature = 14.47 $\pm$ 0.16 °C (range; 11.25–19.45); mean humidity = 91.20 $\pm$ 0.3% (80.6–94.3); mean illuminance = 2.55 $\pm$ 1.8 lux (0–156.05). In only two cases two adults shared the same cave sector, while juveniles did this more frequently

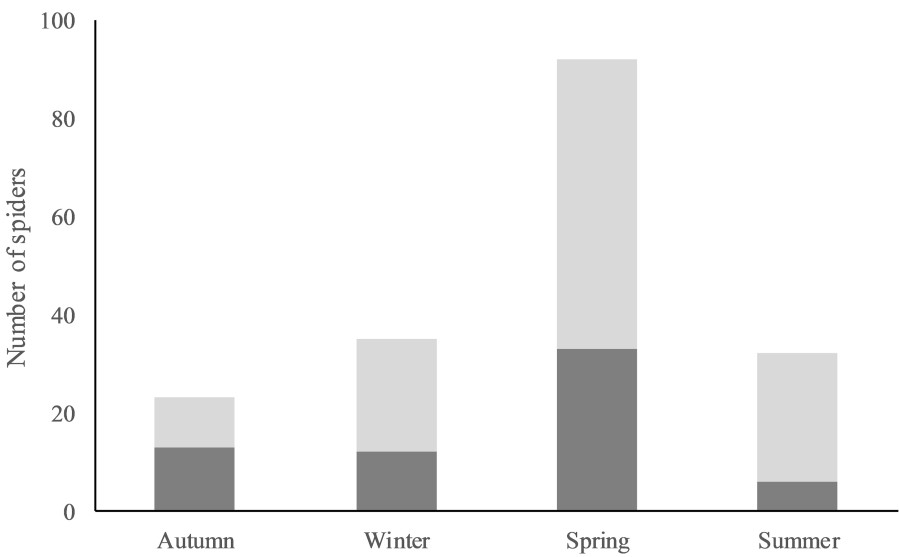

**Figure 2 Observed abundance of *Meta bourneti* spiders in Monte Albo's caves.** Seasonal number of observed spiders is given separating adults (dark grey) and juveniles (light grey) from autumn 2015 to summer 2016.

(four times with an adult and 19 with other juveniles). Two cocoons were observed during autumn, each in a different cave. One of these was observed lying on the ground, already with numerous recently hatched spiders; during winter, spiderlings abandoned the cocoon. No further information on the second cocoon was available.

### Detection probability of *M. bourneti*

In species analysis, the model including depth as covariate was the best model (AICc = 753.38) compared to the other three models (model including season, AICc = 755.72; model including wall heterogeneity, AICc = 756.02; null model, AICc = 756.24). *Meta bourneti* showed an overall low detection probability (0.225). Considering adults only, the model including depth as a covariate was the best (AICc = 383.72) compared to the other three models (model including season, AICc = 389.71; model including wall heterogeneity, AICc = 389.71; null model, AICc = 387.74). Adults showed a very low detection probability (0.108). Finally, for juveniles the model including wall heterogeneity as covariate was the best model (AICc = 559.02) compared to the other three models (model including depth, AICc = 561.78; model including season, AICc = 559.98; null model, AICc = 562.26). The detection probability of juvenile *M. bourneti* was 0.164.

### Spider occurrence

Results of the two analyses (GLMM and GLM) were consistent, thus showing a substantial similarity in the identification of significant variables (Tables 1 and 2). The occurrence of *M. bourneti* was positively related to sector height and humidity. The best GLMM also included the season and the interaction between season and illuminance. Site was also included in the best GLM (Tables 1 and 2). The occurrence of adult spiders was negatively

**Table 1 The best five AICc models relating the presence of *Meta bourneti* (*Meta* spiders, Adults and Juveniles).** In both GLMM and GLM analyses, the presence of the respective studied group was used as a dependent variable. Independent variables were: Height, Width and wall Heterogeneity (Het) of sectors, Season of the survey, mean Temperature (Temp), Humidity (Hum) and Illuminance (Lux) recorded inside each sector. Interactions ($\times$) between season and microclimatic features (temperature, humidity, illuminance) were added as further independent variables. In GLMM analyses both sector and cave identity were used as random factors; in GLMs cave identity was included as an additional independent variable. The "X" indicates the presence of the variable into the respective AICc model; —indicates that the variable was not used in the analyses.

| | Independent variables included in the model | | | | | | | | | | | df | AICc | Δ-AICc | Weight |
|---|---|---|---|---|---|---|---|---|---|---|---|---|---|---|---|
| | Height | Width | Het | Season | Cave | Temp | Hum | Lux | Temp$^{\times}$S | Hum$^{\times}$S | Lux$^{\times}$S | | | | |
| **GLMM** | | | | | | | | | | | | | | | |
| *Meta* spiders | | | | | | | | | | | | | | | |
| | X | | | X | — | | X | X | | | X | 12 | 456.9 | 0 | 0.254 |
| | X | | X | X | — | | X | X | | | X | 13 | 457.3 | 0.41 | 0.207 |
| | X | | | X | — | | X | X | | X | X | 15 | 457.8 | 0.91 | 0.161 |
| | X | X | | X | — | | X | X | | | X | 13 | 458.2 | 1.34 | 0.130 |
| | X | | X | X | — | | X | X | | X | X | 16 | 458.3 | 1.42 | 0.125 |
| Adults | | | | | | | | | | | | | | | |
| | | | | X | — | | | X | | | | 5 | 220 | 0 | 0.220 |
| | | | | X | — | | X | X | | | | 6 | 220.2 | 0.17 | 0.202 |
| | X | | | | — | | | X | | | | 5 | 220.6 | 0.56 | 0.166 |
| | | | | | — | | | X | | | | 4 | 220.9 | 0.89 | 0.141 |
| | X | | X | | — | | | X | | | | 6 | 220.9 | 0.95 | 0.137 |
| Juveniles | | | | | | | | | | | | | | | |
| | X | | | X | — | X | X | X | | | X | 13 | 344.4 | 0 | 0.246 |
| | X | | | X | — | | X | X | | X | X | 15 | 345.1 | 0.74 | 0.171 |
| | X | | | X | — | | X | X | | | X | 12 | 345.2 | 0.81 | 0.164 |
| | X | | | X | — | X | X | | | X | | 12 | 345.4 | 0.96 | 0.153 |
| | X | | | X | — | | X | | | X | | 11 | 345.4 | 0.98 | 0.151 |
| **GLM** | | | | | | | | | | | | | | | |
| *Meta* spiders | | | | | | | | | | | | | | | |
| | X | | | X | X | | X | | | | | 11 | 149.3 | 0 | 0.357 |
| | X | | X | X | X | | X | | | | | 12 | 150.9 | 1.53 | 0.166 |
| | X | X | | X | X | | X | | | | | 12 | 151 | 1.66 | 0.156 |
| | X | | | X | X | | X | X | | | | 12 | 151.3 | 2.02 | 0.130 |
| | X | | | X | X | X | X | | | | | 12 | 151.5 | 2.17 | 0.121 |
| Adults | | | | | | | | | | | | | | | |
| | X | | | X | X | | | X | | | | 11 | 77.3 | 0 | 0.233 |
| | | | X | X | X | | | X | | | | 11 | 77.5 | 0.22 | 0.209 |
| | X | | X | X | X | | | X | | | | 12 | 77.7 | 0.45 | 0.186 |
| | | | | X | X | | | X | | | | 10 | 79.2 | 0.95 | 0.145 |
| | X | | | X | X | | | X | | | | 12 | 78.5 | 1.15 | 0.131 |
| Juveniles | | | | | | | | | | | | | | | |
| | X | | | X | X | | X | | | | | 11 | 102.5 | 0 | 0.315 |
| | X | | | X | X | X | X | | | | | 12 | 103.1 | 0.56 | 0.238 |
| | X | | | X | X | | X | X | | | | 12 | 104.4 | 1.93 | 0.120 |
| | X | X | | X | X | | X | | | | | 12 | 104.5 | 1.96 | 0.118 |
| | X | | X | X | X | | X | | | | | 12 | 105 | 1.99 | 0.117 |

**Table 2  Parameters related to the presence of *Meta bourneti* spiders.** For each group (*Meta* spiders, Adults and Juveniles) the significance of variables included in the relative best AICc model of the respective analysis. Shaded variables are those included in the best model of both GLMM and GLM.

| | | GLMM | | | GLM | | |
| --- | --- | --- | --- | --- | --- | --- | --- |
| | Factor | $\beta$ | $\chi^2$ | P | $\beta$ | $\chi^2$ | P |
| **Meta bourneti** | | | | | | | |
| | Season | | 10.33 | **0.016** | | 4.99 | 0.173 |
| | Cave | | | | | 12.08 | **0.034** |
| | Height | 0.28 | 16.12 | **<0.001** | 0.27 | 17.51 | **<0.001** |
| | Humidity | 13.29 | 13.87 | **<0.001** | 11.23 | 9.64 | **0.002** |
| | Illuminance | −1.71 | 0.01 | 0.917 | | | |
| | Illuminance×Season | | 14.57 | **0.002** | | | |
| **Adults** | | | | | | | |
| | Season | | | | | 0.86 | 0.834 |
| | Cave | | | | | 5.65 | 0.342 |
| | Height | | | | 0.24 | 3.75 | 0.053 |
| | Wall_Irreg | 5.18 | 2.92 | 0.087 | | | |
| | Illuminance | −2.58 | 7.52 | **0.006** | −3.03 | 10.06 | **0.001** |
| **Juveniles** | | | | | | | |
| | Season | | 18.7 | **<0.001** | | 8.9 | **0.031** |
| | Cave | | | | | 14.14 | **0.015** |
| | Height | 0.29 | 14.65 | **<0.001** | 0.28 | 13.73 | **<0.001** |
| | Temperature | 0.34 | 2.89 | 0.089 | | | |
| | Humidity | 17.14 | 16.25 | **<0.001** | 13 | 8.19 | **0.004** |
| | Illuminance | −1.5 | 0.08 | 0.779 | | | |
| | Illuminance×Season | | 10.57 | **0.014** | | | |

related to illuminance (Tables 1 and 2). The occurrence of juvenile spiders was positively related to sector height and humidity; a significant relationship with season was included in the best model of both analyses. The best GLMM also included a significant relationship between season and illuminance, while in the best GLM the site was also included (Tables 1 and 2).

Results of GLM including sector depth as a further independent variable were consistent with those of the previous GLM analyses (Tables S3 and S4).

## Spider abundance

The abundance of *M. bourneti* was related to sector humidity ($F_{1,543.59} = 6.7$, $P = 0.01$) season ($F_{3,566.23} = 3.41$, $P = 0.017$) and the presence of *Hydromantes flavus* ($F_{1,672.34} = 21.91$, $P < 0.001$) and *Oxychilus oppressus* ($F_{1,673.13} = 22.55$, $P < 0.001$). Spiders were more abundant in spring, within cave sectors with high humidity and where *H. flavus* and *O. oppressus* were present. The abundance of adults showed no significant correlation with the variables considered. The abundance of juveniles was related to sector temperature ($F_{1,267.93} = 4.22$, $P = 0.041$), humidity ($F_{1,561.55} = 7.65$, $P = 0.006$), season ($F_{3,580.85} = 4.27$, $P = 0.005$) and the presence of both *H. flavus* ($F_{1,673.15} = 25.65$, $P < 0.001$) and *O. oppressus* ($F_{1,673.59} = 29.73$, $P < 0.001$). Juvenile spiders were generally more abundant in spring,

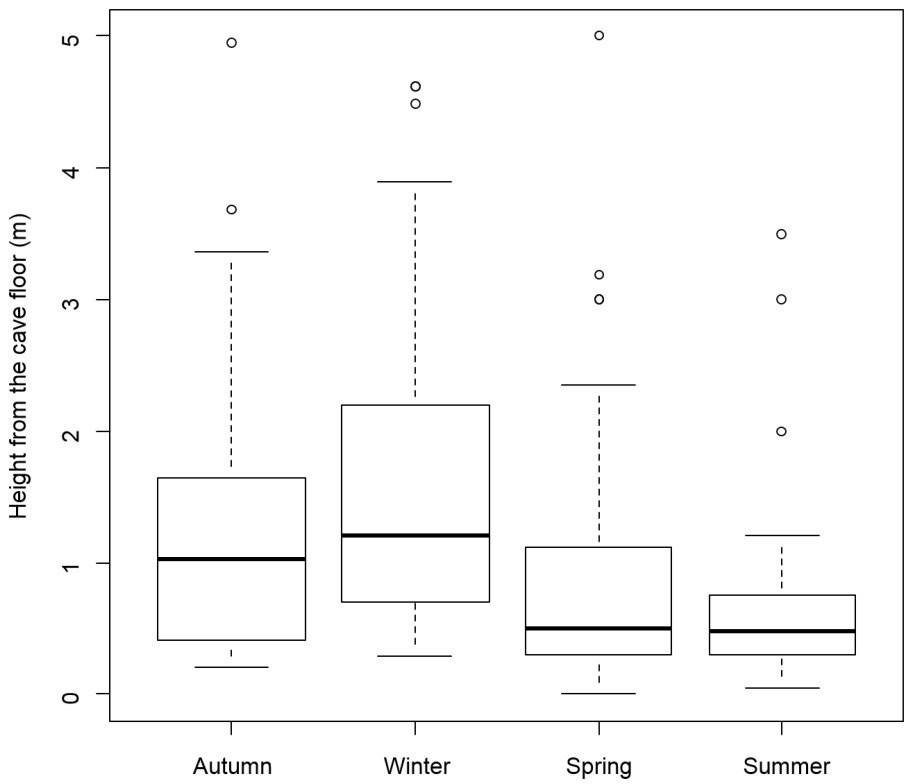

**Figure 3** **Boxplots indicating the vertical distribution of *Meta bourneti* along cave walls.** Differences in vertical distributions of spiders (mean height above the cave floor) among seasons. Horizontal bar inside the box represents the median.

within warm cave sectors showing high humidity and where *H. flavus* and *O. oppressus* were present.

## Spider distribution

Distance from the cave entrance did not differ between age classes ($F_{1,122} = 0.26$, $P = 0.608$) nor between seasons ($F_{3,122} = 0.58$, $P = 0.626$). Vertical distribution of spiders (i.e., height from the cave floor) did not differ between age classes ($F_{1,113} = 0.85$, $P = 0.358$) but a significant effect of season was detected ($F_{3,113} = 6.20$, $P < 0.001$); *Meta* spiders were generally at a lower height during spring and summer (Fig. 3).

## DISCUSSION

*Meta bourneti* spiders represent one of the top predators commonly occurring in Monte Albo caves. Indeed, spiders were present in most of the subterranean environments sampled. The only cave of the dataset in which *M. bourneti* was never observed was located at an elevation exceeding 1,000 m above sea level. Probably, at this high elevation microclimatic conditions are unsuitable for the species (*Lunghi et al., 2018d*; *Mammola & Isaia, 2014*). The largest number of spiders observed occurred in spring, a season in which invertebrates are generally more active (*Bale & Hayward, 2010*). In the populations studied, the life cycle

of *M. bourneti* seems to differ slightly from what was observed in north-western Italian populations (*Mammola & Isaia, 2014*). In September, the cocoon was already spun, and spiderlings started to emigrate in January. This possible variation in breeding phenology probably occurred because the two study areas are characterized by different climatic conditions (data derived from *Hijmans et al., 2005*). It was recently shown that climatic conditions occurring at the surface can significantly influence the subterranean breeding activity of troglophile species (*Lunghi et al., 2018c*). However, the two data collections on *M. bourneti* were performed in different periods (2012–2013 in north-west Italy and 2015–2016 in Sardinia), it is therefore still unclear whether such a divergence may be due to a change in local climate or to an annual fluctuation of climatic conditions. In the future, an improvement in the number of cocoons observed, as well as in repeated surveys over different years, will help in understanding whether populations of *M. bourneti* show divergences in their life cycle.

Detection probability of *M. bourneti* was very low within cave sectors. Besides the low density observed in the studied populations, some other environmental factors may have contributed in lowering spiders detectability (*Nichols, Thomas & Conn, 2008*; *Pollock et al., 2002*). For example, the average ceiling height was usually too high for an exhaustive survey (mean height ($\pm$SD) = 3.19 $\pm$ 2.28 m). Vertical movement of individuals could have put them in a position where they become difficult to detect (*Nichols, Thomas & Conn, 2008*). In addition, another possibility is that the wall heterogeneity sheltered individuals, particularly smaller spiders, from being observed. Despite the general low detection probability, the adopted methodology of data analysis avoided potential biases due to such estimations and highlighted a high consistency of results obtained by both GLMMs and GLMs (Tables 1 and 2). Occurrence of *M. bourneti* was generally related to cave sectors showing high humidity. Sector humidity also positively affected the presence of juvenile spiders, while adults showed a high occurrence in cave sectors with low light (Table 2). These particular microclimatic conditions (high humidity and low illuminance) usually occur in cave areas far from the surface, where external influences are weaker and the microclimate is more stable (*Culver & Pipan, 2009*; *Lunghi, Manenti & Ficetola, 2015*). As was pointed out for both *M. bourneti* and *M. menardi*, these spiders occupy cave areas deep enough to show suitable microclimatic conditions, but still in the proximity of sites with elevated prey abundance (*Lunghi, Manenti & Ficetola, 2017*; *Mammola & Isaia, 2014*; *Manenti, Lunghi & Ficetola, 2015*). However, the tendency of *M. bourneti* to occupy cave sectors with high ceilings is just the opposite of what was observed for *M. menardi* (*Lunghi, Manenti & Ficetola, 2017*). Considering that these two species show similar hunting strategies (*Mammola & Isaia, 2014*), the different preferences of cave sector morphology may be driven by some other ecological factors. For example, in cave sectors with high ceilings, spiders may have more surface (i.e., cave wall) to escape from potential predators present in the same cave sectors (e.g., *Hydromantes* salamanders; *Lunghi et al., 2018b*). Indeed, sector height was particularly significant for juveniles, while for adults this variable was not included in the best AICc model (Tables 1 and 2).

Analyses of spider abundance identified both environmental and biological features as potential determinants. In cave areas with high humidity, *M. bourneti* showed the

highest abundance. Furthermore, the presence of two other species (*Hydromantes flavus* and *Oxychilus oppressus*) had a strong influence on spider abundance. While it is possible that *M. bourneti* shares the same microhabitat preference with these species (*Ficetola et al., 2018*), trophic interactions between *M. bourneti* and these two species may also explain this particular association (*Curry & Yeung, 2013*; *Lunghi et al., 2018b*; *Mammola & Isaia, 2014*). However, compared to *H. flavus*, very limited ecological information on *M. bourneti* and *O. oppressus* are available and thus, future studies are needed to shed light on this particular relationship. Overall, results from spider abundance analyses must be carefully interpreted. The majority of observations were related to juveniles (∼66%) and this may have biased the analysis performed at the species level. Indeed, results from the two analyses (all spiders and juveniles only) were basically the same. When only adults were considered, no significant variables were detected.

No significant differences were found in the horizontal and vertical distribution between age classes. Two spiders were rarely observed inside the same cave sector, and these circumstances generally involved juveniles (Table S2). Information relating to the behavior of this species is virtually absent; hence it is possible that individuals may be territorial, at least in some populations. Considering the limited sample size analyzed here (Table S1), further studies are needed to better comprehend the behavior of *M. bourneti* spiders. Seasonality did not affect *Meta* spiders distribution along the horizontal development of the cave, but it strongly affected the vertical distribution of all individuals (Fig. 3). During hot seasons, spiders were found closer to the cave floor. Air circulation in cave environments is characterized by two main air layers, where the lowest has a cooler temperature (*Badino, 2010*). Therefore, it may be that during hot seasons the temperature of the upper layer becomes too high and spiders move toward the ground floor looking for a more suitable microclimatic condition (*Lunghi, Manenti & Ficetola, 2017*).

## CONCLUSION

This study represents the first analysis performed on island populations of *Meta bourneti*, and was conducted with the aim of adopting a more complete approach to studying the different ecological aspects of these cave-dwelling spiders. *Meta* spiders were found to be widespread in subterranean environments of Monte Albo, but with low densities. The species' life cycle, as well as the distribution of individuals inside caves, appears to be strongly dependent on local climatic conditions, showing some divergences from mainland Italian populations. Microclimate was one of the main features affecting both the presence and abundance of *M. bourneti* in subterranean environments. Morphological cave features may help *Meta* spiders escape unsuitable microclimatic conditions and avoid potential predators. During their subterranean phase, spiders showed the same tendency to avoid the shallowest part of the caves (only one out of 182 observed individuals was found within the first six meters), areas which likely have unsuitable microclimatic conditions. The vertical movement of spiders during different seasons suggests behavior that limits exposure to unsuitable microclimatic conditions. However, further studies on populations

from different geographical regions may help in providing a better overview on the ecology of this widespread cave-dwelling species.

### Funding

The author received no funding for this work.

### Competing Interests

The author declares that he has no competing interests. Natural Oasis is a non-profit association supporting activities like wildlife photography and research. Enrico Lunghi is a member of this local association.

### Author Contributions

- Enrico Lunghi conceived and designed the experiments, performed the experiments, analyzed the data, contributed reagents/materials/analysis tools, prepared figures and/or tables, authored or reviewed drafts of the paper, approved the final draft.

### Data Availability

The raw data are provided in the Supplemental Files.

### Supplemental Information

Supplemental information for this article can be found online at http://dx.doi.org/10.7717/peerj.6049#supplemental-information.

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
