# Peer review of "Ecology and life history of Meta bourneti (Araneae: Tetragnathidae) from Monte Albo (Sardinia, Italy)"

_PeerJ, doi:10.7717/peerj.6049_

## Round 0.1 · original submission · Major Revisions

Dear Enrico,

Thank you for submitting your manuscript to PEER J. I indeed enjoyed reading and learning about Meta bourneti. I apologize for the review taking longer than what is typical for PEER J. Three reviewers commented on your manuscript. Two of the reviewers provide extensive reviews and major revisions will be needed for your manuscript to be considered further for publication. In particular, concern was expressed about reliance on unpublished data. It is standard policy for PEER J submissions that all data be presented and available via open access for reader review. The data needs to be presented in the manuscript or supplemental information and datasets need to be made accessible in on-line data repositories before it can be considered for publication in PEER J.

The manuscript is written in a first person style and I suggest revision in a third person format. I've provided extensive editorial suggestions to help you revise the abstract and introduction. Reviewer one, made some helpful citations about citations and also noted the need for more explanation about the marked difference in detection probability that you note when comparing your results for M. bourneti with that observed for other species.

I encourage you to carefully review their thoughtful suggestions. Please also take note of their suggestions for presentation of the tables and figures.

Thank you again for your submission and I'll look forward to reviewing your revised manuscript. Due to the major revisions required it may require return of the revised manuscript to the reviewers for a second look.

Best regards

Don't hesitate to contact me if you have questions.

Jay

·

Basic reporting

The author investigated the auto-ecological requirements and the seasonal and spatial dynamics of the cave-dwelling spider Meta bourneti, using a few subterranean habitats as model sites. Overall, I found the study interesting and useful in providing novel ecological informations about this widespread mediterranean species.

I enclosed below a list of suggestions for improvements, which can be quickly addressed. I think that the referenced papers concerning the biology of Meta spiders are in a few places slightly incomplete or ambiguous. I have therefore provided several suggestions in this sense.

As I am not mother tongue, I have not evaluated the English.

>>>> Line comments:

--- Lines 16-17 (abstract): I suggest rephrasing to "[...] most common cave predators occurring in the Mediterranean basin". In continental and alpine European areas, the species is actually very rare, if not absent at all.

--- Line 17: change "orb web" to "orb-weaver"

--- Line 17 (abstract): after this first mention of the model species name, could be useful to add order and family, i.e. "Meta bourneti (Araneae: Tetragnathidae)." Also in the introduction.

--- Lines 54-58: please add a reference(s) for the cave zonation.

--- Lines 61-63: I slightly disagree. The "degree of adaptation to cave life" is mostly referred to as 'troglomorphism' (after Christiansen 1962). Conversely, the references herein cited (Pavan, Novak et al. and Sket) are about the ecological classification of subterranean organisms based on the preferential habitat in which these species perform their life cycle. Accordingly, I suggest to change the wording to: "A species’ degree of association to subterranean conditions represents the base of the general ecological classification used in distinguishing between different groups of cave-dwelling species . . . " or something similar.

Christiansen K (1962) Proposition pour la classification des animaux cavernicoles. Spelunca 2: 76–78.

--- Line 72: indeed, the classification is very limited and debated. An interesting recent paper dealing with this topic is also:

Trajano E, Carvalho MR (2017) Towards a biologically meaningful classification of subterranean organisms: a critical analysis of the Schiner-Racovitza system from a historical perspective, difficulties of its application and implications for conservation. Subterranean Biology 22: 1-26.

--- Lines 76-77: I agree. Nevertheless, I guess that it is not informative and entirely correct to cite a sample of random studies focused on very specific organisms (crickets: Studier et al., 1986; woodlouse: Fernandes, Batalha & Bichuette, 2016; salamanders: Lunghi et al., 2018e) or subterranean systems (show caves: de Freitas, 2010) when referring to ". . . the undeniable increase of interest in underground ecological spaces and related biodiversity which has occurred in the last decades . . . ". This is because local studies, by definition, fail in showing that there has been an increased interest in subterranean biology as a discipline; hey just represent very specific cases. Possibly it would be more appropriate to cite a sample of important books and review papers, as these are usually more ample in their scope and deal with a large number of subterranean systems/organisms/studies and often consider an historical perspective. Among others, Culver & Pipan (2009), Culver & Pipan (2014, Already cited), Juan et al. (2010), Gibert and Deharveng (2002), Chamaco (1992) would be appropriate references.

Camacho, A. I. (1992). The natural history of biospeleology. Monographs of the National Museum of Natural Sciences.

Culver, D. C., & Pipan, T. (2009). The biology of caves and other subter- ranean habitats. Oxford, UK: Oxford University Press.

Gibert, J., & Dehrveng, L. (2002). Subterranean ecosystems: A truncated functional biodiversity. BioScience, 52, 473–481.

Juan C, Guzik MT, Jaume D, Cooper SJB (2010) Evolution in caves: Darwin’s ‘wrecks of ancient life’ in the molecular era. Molecular Ecology 19: 3865–3880.

--- Lines 82-85: The life cycle of Meta menardi was firstly studied by Peter Smithers in two independent experiments, which should be mentioned here.
Later on, Chiavazzo et al. (already cited) and Ficetola et al (2012) provided evidences about the mechanisms enhancing the migration of the spiderlings form the cave and a potential clue used by the spiderling to colonize new caves, respectively.

Conversely, although being a very interesting study, as far as I remember Manenti et al. (2015) focused onh the microhabitat and trophic preference of M. menardi, rather than on the life cycle. Thus I would remove this reference in this particular place (but I may be wrong).

For a schematic representation of the life cycle, you may also refer to Figure 6 in Mammola & Isaia 2014.

Ficetola F.G., Pennati R. & Manenti R., 2012 - Do cave salamanders occur randomly in cavities? An analysis with Hydromantes strinatii. Amphibia-Reptilia, 33: 251-259

Smithers P., 2005 - The early life history and dispersal of the cave spider Meta menardi (Latreille, 1804) (Araneae: Tetragnathidae). - Bulletin of the British Arachnological Society, 13(6): 213-216.

Smithers P. & Smith F.M., 1998 - Observations on the behaviour of second instars of the cave spider Meta menardi (Latreille, 1804). Newsletter of the British Arachnological Society, 81: 4-5.

--- Lines 88: As far as I know, none of the authors cited, but Smithers 2005, made direct observations on the hunting strategy of Meta menardi. I guess that Tearcafs (1972) was the first authors to speak about this peculiar mode of hunting.

Tercafs, R., 1972. Biométrie spatiale dans l’écosystème souterraine: repartition du Meta menardi Latr. (Argiopidae). Int. J. Speleol. 4, 351e355.

--- Line 89: You may also cite Pastorelli and Laghi here.

--- Line 93: Concerning the overview on biological studies focused on Meta menardi, you may also include this seminal paper:

Ecker R. & Moritz M., 1992 - Meta menardi (Latr.) and Meta merianae (Scop.): On the biology and habitat of the commonest spiders in caves of the Harz, the Kyffhauser, Thuringa and the Zittau mountains. Mitteilungen aus dem Zoologischen Museum Berlin, 68 (2): 345-350.

as well as a recent paper focused on the histology:

Lipovšek, S., Leitinger, G., Novak, T., Janžekovič, F., Gorgoń, S., Kamińska, K., & Rost-Roszkowska, M. (2018). Changes in the midgut cells in the European cave spider, Meta menardi, during starvation in spring and autumn. Histochemistry and cell biology,149(3), 245-260.

--- Lines 102-105: You may also specify that this study is the first of his kind set in an area where only Meta bourneti is present. Thus, by design of the study, you controlled for the effect of interspecific competition with M. menardi. You mentioned this later at
Lines 116-118, but it may be useful to mention this earlier in order to focus the reader' attention on one of the most important novelties of this study.

--- Line 125: "M. bourneti" instead of "Meta bourneti"

--- Line 211: First time you mention the fact that you recorded the presence of cocoons. Perhaps add it in the methods at line 125.

--- Line 264: "There" is repeated twice in the sentence.

--- Line 270: Hijmans et al. provided climatic rasters for the whole globe. Citing this paper like this is imprecise. I would rephrased as such: ".... because the two study areas are characterized by different climatic conditions (data derived from Hijmans et al., 2005).

--- Lines 279-280: "far from the surface" instead of "far from the connection to the surface"?

--- Line 304 (and elsewhere): the term "subterranean" is in general more adequate than “underground”.

--- Line 328 ". . . the shallowest part of the caves . . ." Can you provide the spatial interval, e.g. 1–5 m from the entrance?

Experimental design

>>> Scope and relevance

This research appears to be within the scope of PeerJ, and contribute to fill a knowledge gap in the biology of Meta bourneti. However, the aims of the study are not clearly given in the last paragraph of the introduction and are therefore slightly vague ("provide first information related to the ecology and life history"). I think the questions addressed here are:
1) the abiotic and biotic factors affecting spatial and temporal distribution of the spiders within the caves;
2) to see whether there are difference in these patterns depending on the stage of the spider (juveniles vs adults)
3) to gather and summarize informations about the life history of the species (e.g. the observations made on cocoons).
Can you please recast this paragraph to provide a list of specific aims or working hypotheses? It would be really helpful to give more structure to the paper and support your findings.

>>> Sampling methodology

The paper is based on a dataset with information on Meta bourneti and inhabited caves from the Monte Albo (north-east Sardinia, Italy), which is currently unpublished (Lunghi et al., unpublished). Unless the dataset will be published prior to this paper, more informations about this dataset must be included in the method section. Useful information to report would be:

- name, cadaster, coordinates and general informations about the sampled caves (e.g. one table providing informations about the sites?)
- minimum and maximum cross distance between caves
- number of sampling sectors/cave (I guess it was variable depending on the planimetric development of the specific site)
If convenient, you may also consider making a map of the study area.

--- You gathered several abiotic parameters within the cave: maximum height and width, wall heterogeneity, average temperature (°C), humidity (%) and illuminance (lux). Which probes and methods did you used for acquiring these data?

>>> Statistics

The data are well analyzed using relevant statistical techniques. The statistical methods are in general well described, but I have a few doubts/comments/suggestions:

--- I have doubts about the detection probability analysis and relative results, which should be clarified upon acceptance. [Disclosure: I am not expert in detection probability analyses, and I have never performed similar analyses myself. So, take my comments as observations made by a non-expert]

In my experience, Meta spiders are very easily detected in caves. Unlike other cave-dwelling spiders, especially Linyphiidae, Leptonetidae and Nesticidae, they:
- are very big;
- mostly occur in the superficial and illuminated cave sectors;
- rarely dwell in deep, unaccessible crevices (This is mainly because they need a lot of 3-dimensional space for web spinning).

Accordingly, I thought that the detection probability of Meta spiders would be in general very high, i.e. close to 1. Indeed, in a similar study by Manenti et al. (2015), detection probability of M. menardi was found to be as high as 97%.

Therefore, I found it very strange that in this study, you had an overall detection probability of 0.2. Are you sure of this result? Isn't it that the species just occurs at very low density within the study area (Lines 320-322)?
Most importantly, with such a low detection probability, can you trust the other analyses? If this result is correct (which I personally doubt), you are severely underestimating the abundance and presence/absence of these spiders in your sampling sectors, thus making inferences based on a biased dataset.
Surprisingly, this result is not further addressed in the paper, and should be at least commented in the discussion.

--- Concerning the covariates influencing the detection probability:
Why not considering also the developmental stage of the spiders (juveniles versus adults)? I think that, in general, larger spider should be more easily detected. I also think that wall heterogeneity should be included, as the presence of wall irregularities may offer micro-niches which may hide some individuals, thus preventing their detection at a superficial and quick inspection.

--- Lines 137-138. It would be useful to provide the reference of each individual R package just after its mention, and not as a list at the end of the sentence.

--- Line 157. What was the proportion of zeros and ones in the binomial dataset. Was it balanced? Please specify.

--- Line 178. Why adding a random factor was not possible? Please specify.

--- Lines 223-224. Please reconsider. In the model concerning juveniles, you have a delta AIC of less than 2 among the three models. Therefore, you cannot tell apart which one is best (see, e.g., Zuur et al. 2009).

--- Line 257: please specify the reference category.

--- Table 2. replace B with the greek letter Beta.

--- Figure 1. This figure is very blurry in the pdf, please try uploading one at higher resolution.

Validity of the findings

Although local in scale, the study is certainly useful to "[...] better comprehend the ecology of these widespread cave-dwelling spiders" (Lines 42-43). This is because very little ecological information is available regarding this poorly studied, yet widespread, species. In this sense, this research will certainly make a useful contribution.

I only have a major doubt about the detection probability analysis. I would like to see it discussed in more details before acceptance.

·

Basic reporting

This ms provides a survey concerning ecology and life history of Meta bourneti a troglophile spiders from the caves of Monte Albo (Italy). The paper is good in content, with good drawings and photos and is recommended for publishing. My remarks and recommendation are pointed mainly to, technical omissions, marked in the text (by track change).
I can not check the quality of the English language, because I am not native speaking.

Experimental design

no comment

Validity of the findings

no comment

Additional comments

My remarks and recommendation are pointed mainly to, technical omissions, marked in the text (by track change).

·

Basic reporting

The majority of the article was very well written but there were several errors in the choice of English words or phrases. I made note of these small errors directly on the PDF version of the manuscript. Two general comments I would make are that 1) the first use of the species name in the manuscript should be Meta bourneti Simon, 1922. After the first use of the full species name use only M. bourneti later in the manuscript. 2) Using the word model after GLM and GLMM is redundant because the word “model” is part of the acronyms for Generalized Linear Models and Generalized Linear Mixed Models.
I felt that the introduction did a good job of providing a sufficient background for the reader to understand the importance of the subject matter and how the current study can provide useful information to fill a gap in the current level of understanding about organisms in underground ecosystems.
I am concerned about the reliance on unpublished data throughout the manuscript. I am slightly confused by this because there is a large amount of raw data provided in the supplemental information and therefore it should not be cited as “unpublished data”. I would recommend that the author simply make all of the necessary data available in this current study and therefore there is no issue with relying on unpublished data. In addition to the raw data, it would be helpful to have summary tables of the supplementary information. These tables are cited in the text of the manuscript but I only saw Table S3; for Table S1 and Table S2 there was only raw data provided. In the PDF document I made specific comments about the figures. I feel that Figure 1 should be changed to simple black/gray colorations and that the title and legend should be removed because this information is already included in the figure description. The figure description should also be placed underneath the figure. I am not sure if Figure 2 is necessary because it focuses on the observations of only two cocoons that were found during the study and I do not feel as though this provides sufficient data and it does not appear to be important for the overall objective of the study. I feel that Figure 3 is also unnecessary because it only focuses on the non-significant differences between spider age classes for the distance from the cave entrance and the height above the cave floor. These results are stated clearly in the results of the manuscript and I do not feel it is necessary to provide a figure for these results. The author stated that there was significant differences between seasons but this was not shown in the figure. I would recommend changing the figure to include the factor of season. In the new figure the description should be placed under the figure and it should contain the statistical test information and results.
In the PDF I made a couple comments about the relevance for two components of the results, in which I don’t feel were relevant to the main objective of the study. 1) I feel that the observations on the two cocoons found is not sufficient enough information to be included in the study and is not necessarily relevant to the main objective of the study. 2) Another aspect of the results I feel should be better explained is the correlation between spider presence with the presence of Oxychilus oppressus. It was mentioned that it could be related to predator-prey interactions but I am not sure how this snail species is relevant to the food-web interactions of M. bourneti.

Experimental design

This study represents original primary research that meets the aims and scope of the journal. The research question was well defined, relevant and meaningful. The introduction did a good job of stating how the study fills an identified knowledge gap about underground ecosystems. I feel that the data used in this study was the result of a rigorous investigation on underground ecosystems around Monte Albo in Sardinia, Italy and that it was conducted to a high technical and ethical standard. I feel that the majority of the methods were described with sufficient detail and clarity to be replicated. However, as I stated earlier, I have concerns about how the analyses are based off of the “unpublished data”. This could be easily fixed by providing all of the necessary information as supplementary data, which could have been done here, but it is somewhat difficult to tell in the way that the raw data was organized in excel files; with no summary tables for the supplementary data.

Validity of the findings

The data used in this study appears to be sufficiently robust and was analyzed in a statistically sound and controlled manner. I will just reemphasize that I feel the author should address the issue of using unpublished data as the basis for the study. Also, the supplementary data should be provided in a manner that is easier to review by including summary tables that focus on the data used for the analyses in the manuscript. The majority of the conclusions are well stated and are limited to supporting results; the only exceptions to this would be the author’s conclusions about the observations on the cacoon observations, the importance of trophic interactions with Oxychilus oppressus and the correlation with height distributions and predator avoidance (I included comments on these points in the PDF). I feel that it would be beneficial to provide more supportive evidence and/or explanation on these three aspects included in the conclusions section of the study.

---

## Round 0.2 · Minor Revisions

Enrico,

I enjoyed learning about the cave spiders of Sardinia described in your manuscript : Ecology and Life History of Meta bourneti (Araneae: Tetragrnathidiae) from Monte Albo (Sardinia, Italy), As the academic editor of your article I feel it will be suitable for publication in Peer J after you have made some revisions. I've edited your manuscript and have incorporated many of the changes suggested by Reviewer 3. Please don't be concerned by the numerous suggested changes, many were made to convert what was a single sentence with a semi-colon into two sentences. There were also tense changes and some reorganization of wording. Please be sure that your intended meaning has not been changed by any of the suggested changes. There were also some questions posed that will necessitate reworking some of the narrative. Please be sure to read through and adhere to the suggestions of the two reviewers. In addition, please consider two changes in particular suggested by reviewer 3. Figure 1 does need to be changed to an image with more clarity. In addition, the reviewer has indicated that the supplemental data was not accessible in the format provided. Please adhere to the journals data formatting requirements. I have asked that the journal also relay an MS Word document to you that should make your editorial changes easier. You can accept the changes and then respond to the questions posed. Don't hesitate to contact me if you have questions. We will look forward to receiving your revised manuscript. Thank you for considering Peer J for you submission.

Jay

·

Basic reporting

I've read the rebuttal letter and the revised ms. I'm overall satisfied by the way in which my comments and suggestions were handled. I also thank the author for the detailed clarifications on the detection probability phylosophy and relative analyses.

Find below a few final minor remarks, and congratulation for this nice contribution.

L19: perhaps remove "full"

L30-31: please add taxonomical authority and (class:order) for Hydromantes flavus and Oxichilus oppressus

L106: add a comma before "and": i.e. ... occur, and iv ...

L138: also here, please give authorities for new species herein introduced for the first time

L139-140: I agree for most of the species listed here. However, probably adult Tegenaria does not interact with Meta in that sense...

L232 and 235: "null model" instead of "model without covariates"?

L276: Spell "Meta" in full at the beginning of the discussion

Experimental design

N/A

Validity of the findings

N/A

·

Basic reporting

In the manuscript, “Ecology and life history of Meta bourneti (Araneae: Tetragnathidae) from Monte Albo (Sardinia, Italy)”, the author describes the abundance and ecological parameters of M. bourneti spiders that are commonly found in cave systems in the Mediterranean region.
I believe the manuscript provides interesting new information on rarely studied cave ecosystems and a species of spider that could be important to the food web interactions in these systems. I can see that the author put substantial effort into making changes according to the previous comments made by reviewers, but I believe there are still considerable amount of changes that need to be made before it is ready for publication.
One of the major concerns I have is that I believe it needs to be edited closely by a native English speaker due to some consistent errors in the grammar and sentence structure. I have gone through the manuscript and made changes and offered comments on the PDF version of the manuscript to help correct these commonly found errors but some of the sentence structure made it difficult to read at times (e.g., Lines 123-127 and Lines 337-338).
I appreciate the author providing the raw data in the supplementary information (raw-daa-table_S2 and Table_S1_dataset) but when I open the Excel (CSV) file, it is difficult to review the data because it is not organized in a way that can be easily understood. I am not sure if this is due to the process of uploading the data to the PeerJ review system but I believe it is because it was copy and pasted as a .txt file directly into Excel and therefore the data is not organized into individual columns for each of the measured variables. All tables should have similar formatting, including the tables in the supplementary materials. The tables in the supplementary materials are also missing descriptions. I have made some changes and comments to Tables 1 and 2 in the PDF of the manuscript and I would suggest that the tables in the supplementary materials be edited to fit the changes made to Tables 1 and 2.
I made some minor edits and comments to Figure 2 and 3. I think the addition of the map in Figure 1 could be beneficial but the resolution of the image is very poor and it will need to be improved if the author wants to include it.

Experimental design

I feel that the author made the necessary changes in describing the methods used in this study in the revised version of the manuscript and I do not have any more comments for this section

Validity of the findings

As I mentioned previously, I feel that the raw data provided in the supplementary information could be provided in a way that is easier to review in the excel file format. But apart from this, I feel the author put sufficient effort in providing the necessary data that was listed as "unpublished data" in the first version of the manuscript.

---

## Round 0.3 · accepted · Accept

Thank you for your submission to PeerJ and your careful revision of your manuscript.

#